

# Effect of control measures on the pattern of COVID-19 Epidemics in Japan

Tomokazu Konishi

Graduate School of Bioresource Sciences, Akita Prefectural University, Akita, Japan

## ABSTRACT

**Background:** COVID-19 has spread worldwide since its emergence in 2019.
In contrast to many other countries with epidemics, Japan differed in that it avoided lockdowns and instead asked people for self-control. A travel campaign was conducted with a sizable budget, but the number of PCR tests was severely limited. These choices may have influenced the course of the epidemic.

**Methods:** The increase or decrease in the classes of SARS-CoV-2 variants was estimated by analyzing the published sequences with an objective multivariate analysis. This approach observes the samples in multiple directions, digesting complex differences into simpler forms. The results were compared over time with the number of confirmed cases, PCR tests, and overseas visitors. The kinetics of infection were analyzed using the logarithmic growth rate.

**Results:** The declared states of emergency failed to alter the movement of the growth rate. Three epidemic peaks were caused by domestically mutated variants. In other countries, there are few cases in which multiple variants have peaked. However, due to the relaxation of immigration restrictions, several infective variants have been imported from abroad and are currently competing for expansion, creating the fourth peak. By April 2021, these foreign variants exceeded 80%. The chaotic situation in Japan will continue for some time, in part because no effort has been made to identify asymptomatic carriers, and details of the vaccination program are undecided.

## INTRODUCTION

The worldwide COVID-19 pandemic continues as variants mutate rapidly (*Konishi, 2021c*). In Japan, the epidemic began with the spread from a cruise ship. Hitherto, there have been four peaks. As of June 2021, COVID-19 has not been contained, people living in major cities have been asked to stay at home voluntarily, and travel to Japan from abroad is restricted.

The responses to COVID-19 in Japan were different from those of some countries that succeeded in controlling the disease. Strict lockdowns were not carried out; instead, Japan issued states of emergency three times, urged the public to refrain from going out unnecessarily and eating out late at night, and cancelled events. These measures were

Corresponding author
Tomokazu Konishi,
konishi@akita-pu.ac.jp

merely a request for self-restraint, and there were no confirmations, penalties, or compensation. Schools were closed only during the first state of emergency. In the midst of the second peak, the government launched a travel-promoting campaign, "Go To travel," which provides residents with subsidies of up to 50% on transportation, hotels, restaurants, and shopping; 26 billion dollars was proposed for the budget to do so. However, the number of public polymerase chain reaction (PCR) tests is fairly low, and testing is only performed on patients with obvious symptoms. Initially, the plan was to find clusters and infection routes, but this scope was beyond the capacity of the investigation agency and has already been substantively abandoned. The number of small vendors performing PCR tests is increasing, but the results are not counted in official records.

Immigration procedures have often been criticized for being loose (*Edamatsu, Shimoji & Satou, 2021*). Upon arrival, infection is checked *via* PCR or antigen testing. Visitors are then asked to stay in one place for two weeks, but this is not compulsory, and 20% became uncontactable.

The same laissez-faire policy is applied to domestic PCR-positive patients. There is an insufficient number of facilities in which to isolate mild and asymptomatic patients. As of May 15, 2,306 people in Tokyo (*Tokyo Metropolitan Government, 2021*) and 13,499 people in Osaka (*Osaka Prefecture, 2021*) were asked to stay at home or in hotels. However, this request for self-restraint was not compulsory.

Medical resources have been exhausted. The patients who were staying home (or in a hotel) were not administered medicine or oxygen. Among them, 1,340 people in Tokyo and 2,799 people in Osaka have requested hospitalization, but 122 people have died without being hospitalized as far as the police have acknowledged (*NHK, 2021a*). From January 6–28, 110–90% of the beds for severely ill patients in Tokyo hospitals were occupied, and on April 21, and 80% of such beds were occupied in Osaka hospitals (*NHK, 2021b*). At the end of April, of the 13 calls for emergency services, eight were turned down in an emergency hospital in Osaka, which is a highly unusual situation (*Inoue & Tanabe, 2021*). In this study, we examined the situation of epidemics with respect to changes in the relative proportions of virus variants as well as kinetics.

All sequence data available in GISAID (*Elbe & Buckland-Merrett, 2017*) were observed in an objective manner by considering the sequence matrix as a multivariate variable and applying principal component analysis (PCA; *Jolliffe, 2002*). This approach is different from phylogenetic trees (*Yang & Rannala, 2012*), which require many unverifiable assumptions that reduce the objectivity of the analysis (*Ellis & Silk, 2014*). Because several directions of differences can be observed individually, fine classification can be performed with high reproducibility. These data were observed in chronological order and compared with the number of confirmed cases, the number of overseas visitors, and the number of PCR tests performed. It should be noted that a tree does not produce data in a format that can be compared to other information. Therefore, it is difficult to integrate time-course studies with a phylogenetic approach. Additionally, a kinetic approach was applied using the logarithmic growth rate.

## MATERIALS & METHODS

### Data source

Nucleotide sequences of 15,746 samples in Japan were obtained from GISAID (*Elbe & Buckland-Merrett, 2017*) on May 5, 2021. Sequences were aligned using DECIPHER (*Wright, 2015*), which uses secondary structure predictions in the process. Because PCA is not sensitive to differences caused by alignment methods or to the parameter conditions of calculations (*Konishi et al., 2019*), DECIPHER was selected for its faster calculations with a large number of sequences. The aligned sequences were summarized by PCA, as explained in the next section, and then the PCs were compared with the collection date of the sample. All calculations were performed using R (*R Core Team, 2020*). The ID, acknowledgements, and scaled PC (sPC) of the samples and the sPC of the bases are available from Figshare (*Konishi, 2021b*). The number of confirmed cases and the number of PCR tests were obtained from the Ministry of Health, Labour and Welfare, Japan (*Ministry of Health Labour & Welfare, Japan, 2021a*). Those of Tokyo were from the home page of Tokyo City (*Tokyo Metropolitan Government, 2021*). The number of foreign visitors was obtained from an official statistics counter (*e-Stat, 2021*). The number of current cases and PCR tests in the other countries were obtained from the official homepage of the Ministry of Health of the corresponding countries. The nomenclatures of the variants are based on the Pango Lineage in GISAID, version 2021-05-27 (*Rambaut et al., 2020*).

### PCA

This analysis represents the differences among samples of multivariate data through a set of common directions, which are shown as independent vectors (*Jolliffe, 2002*). Here, the samples were mutations that evolved from the original virus. Hence, the samples will fall into several related groups; each group is different from the others, with a unique direction common to the group. The sequence matrix is converted to a stack of Boolean vectors to allow for calculation, replacing specific positions from 0 to 1 for the corresponding bases (*Konishi et al., 2019*). When $m$ is the number of samples (in reality, to cover sequences with $n$ bases, the length of a Boolean vector becomes $5n$, *i.e.*, A, T, G, C, and (-); here, for simplicity, it is described as below), the matrix $X$ is given as follows:

$$X = \begin{pmatrix} 01_{11} & \cdots & 01_{1n} \\ \vdots & \ddots & \vdots \\ 01_{m1} & \cdots & 01_{mn} \end{pmatrix}.$$

Next, the average of the samples, $a$, is found, and $X$ is centered by subtracting the average from each row: $a = (a_1, a_2, \cdots a_n)$, $C = X - a$. This centered matrix, $C$, was applied to the PCA. It is subjected to singular value decomposition, $C = U\Sigma V^*$, where $U$ and $V$ are unitary matrices that specify the directions of the differences. According to the definition of the unitary matrix, $V^*V = I$ and $VV^* = I$, the scale of each column and row is one, $V^*V = I$, and $VV^* = I$. Each of their columns can be regarded as vectors that
convert $C$ to be represented on a specific axis. $\Sigma$ is a diagonal matrix that records the scaling of each axis in descending order.

The PCs for the samples, $S$, were found to be $S = CV = U\Sigma$. The $CV$ indicates the rotation of $C$ around the central origin, retaining the shape of $C$. The results of the rotation are along the axes. This is the same as $U\Sigma$, which is a unitary matrix given the scale. Each of the columns of $S$ represents the PCs: the leftmost column is PC1, and the second column is PC2. The descending character of $\Sigma$ orders the scaling of PCs. All of the information is conserved, and all calculation steps are reversible. Once found, axis $V$ can be applied to other sets of matrices (*Konishi, 2015*). This characteristic is beneficial when applying classification to newly found samples. Here, the axes used were found by using worldwide samples.

PCs for bases, $B$, can be found in the diagonal direction of the above, $C^* = V\Sigma U^*$, as $B = C^*U = V\Sigma$. Therefore, $S$ and $B$ are inextricably linked; for example, samples with many positive $B$ in an axis will become highly positive in $S$ along the same axis. In contrast, a sample's characteristics that show a high score on an axis will appear on the same axis as $B$. This characteristic is beneficial for identifying new variants and their corresponding mutations. A mutation(s) characteristic of a group of samples gives a particular $S$ value along an axis. The mutation(s) appear in $B$.

To enable comparisons with datasets with different sizes of $n$ or $m$, PCs can be scaled to different sizes (*Konishi, 2015*). The scaled versions of $S$ and $B$, sPC for samples and sPC for bases, are $S/\sqrt{n}$ and $B/\sqrt{m}$, respectively. These scaled versions were used in this study.

For these averages $a$ and axes $V$, we used a sequence matrix collected evenly from all over the world as of October 2020 (*Konishi, 2021c*) to allow for comparison of the variants that are prevalent around the world with those prevalent in Japan. The axes represent the changes that have occurred in some countries, including Japan. If a change occurred in Japan, a specific value would appear on the same axis. In addition, to observe newer variants, a new set of axes at May 2021 was also used. Both sets of axes can be downloaded from Figshare (*Konishi, 2021b*).

The conditions for finding variants that produced peaks in each country were as follows (*Konishi, 2021a*):

(original axes)
group 0: −0.004 < PC1 < 0.004, PC2 < 0.004
B.1.177: PC62 > 0.0035, PC63 > 0.009
group 2: PC2 > 0.004
B.1.160: PC118 > 0.002
B.1.2: PC28 > 0.01
(new axes)
B.1.1.7: PC1 < −0.016
B.1.351: PC23 > 0.006 & PC25 > 0.006
B.1.617.1: PC56 > 0.004 & PC60 > 0.001
B.1: PC21 > 0.0024
Table 1 **The domestic variants.** Colours are those indicated in Fig. 4A and 3D. The characteristics in PCA used for the definition, peak date, and the mutated amino acids are presented.

| Colour | Characteristics | Peak | Residue |
|---|---|---|---|
| Tangerine | (−) B.1.1.214 | 2020/04/13 | (−) |
| Water blue | PC192 > 0.00034 B.1.1.214 | 2020/08/03 | 1a.1361:S/P; 1a.3371:P/S; N.414:A/V; N.151:P/L |
| Green | PC66 > 0.00034 B.1.1.214 | 2021/01/04 | 1b.1567:P/L; 1b.2684:R/I; N.90:A/T; N.234:M/I; |
| Blown | PC114 < −0.00071 B.1.1.214 | 2021/01/04 | 1b.1567:P/L; 1b.2684:R/I; N.234:M/I |
| Purple | PC78 > 0.0013 B.1.1.214 | 2021/01/04 | 1a.110:H/R; 1a.3847:V/I; 1b.1567:P/L; 1b.2684:R/I; E.72:L/I; 6.22:F/L; N.234:M/I; |
| Flesh pink | new PC5 < −0.01 B.1.1.215 | 2021/01/10 | 1a.727:T/I; 1a.2039:L/F; 1a.4348:K/R; 1b.1567:P/L; 1 b.2684:R/I; S.675:Q/H; N.234:M/I |
| Orange | PC171 > 0.00088 B.1.1.214 | 2021/01/11 | 1a.2702:Q/H; 1a.2981:S/F; 1b.1567:P/L; 1b.2684:R/I; S.720:I/V; N.234:M/I; |
| Coral red | new PC15 < −0.009 PC66 < 0.00034 B.1.1.214 | 2021/01/16 | 1a.2259:M/I; 1a.3722:Y/C; 1b.1567:P/L; 1b.2143:A/S; 1b.2684:R/I; S.261:G/V; S.675:Q/H; N.234:M/I |
| Blue green | PC30 < −0.002 R.1 | 2021/03/29 | 1b.1362:G/R; 1b.1936:P/H; S.152:W/L; S.484:E/K; S.769:G/V; M.28:F/L; N.1-101:M/-; N.187:S/L; N.418:Q/H |

The variants that arose in Japan were found with the help of K-Means (*Hartigan & Wong, 1979*), which is a non-hierarchical method that classifies samples into a given number of groups. Samples were divided into eight groups, and for each group, each PC axis was compared to the collection date. The number of groups, eight, has an arbitrary nature. It was chosen since it was the largest number, after repeating this process several times, that did not produce small groups that were almost worthless. Although K-Means is a method with low reproducibility, we selected the trials where the grouping did not differ greatly from other trials as a result of the repetition. This arbitrariness is an inevitable consequence of the classifications as it is arbitrary which mutations are considered as independent variants. Samples that showed a specific value in an axis (or multiple axes) at a specific time period were taken as new variants, and the PC value was used to define it (rather than the results of K-Means). This process was repeated several times, including by using the new axes, and a total of 10 variants were identified. Each variant was compared to the variant that produced the first peak (shown in tangerine) to determine which amino acids had changed (Table 1).

## Kinetics

When the infection is early and there are many uninfected people, we can use a simple model (*Wallinga & Lipsitch, 2007*) in which the number of confirmed cases on the $n$th day, $c_n$, increases or decreases exponentially. Hence, the logarithmic growth rate $K$ was calculated to observe the fluctuation of cases in more detail. When a variant grows exponentially, $K$ becomes constant, and $c_n$ increases with constant doubling time,

$$d = 1/K, \; c_n = c_0 2^{n/d} = c_0 2^{Kn}. \tag{1}$$

Taking the logarithm of both sides gives the following formula: $\log_2 c_n = Kn + \log_2 c_0$. Differentiating these two sides by $n$, we find $K$ as $d\log_2 c_n/dn = K$.

Hence, changes in $K$ at any $n$ were estimated as $K_n = (\log_2(c_{n+7}/c_n))/7$; the 7-day difference was set to minimize the fluctuations due to the day of the week. Because the base was set to 2 here, the inverse of $K$ shows the doubling time; for example, if $K$ is 0.1, the doubling time is 10 days.

For convenience, we also included the more popular basic reproduction number $R_0$. Here, the infection period of $\tau$ is assumed to be constant at 5 days (*Alene et al., 2021*). Eq. (1) can be transformed using $R_0$ and $\tau$ by approximating the event based on the idea that an average carrier infects $R_0$ people on day $\tau$ of infection:

$$c_n = c_0 R_0{}^{n/\tau} \tag{2}$$

From Eqs. (1) and (2), we get $R_0 = 2^{K\tau}$.

Because of sampling errors, the estimations of $c_n$, $K$, and $R_0$ fluctuated; they were smoothed by a *locally weighted scatter plot smooth* function, which finds the center of data by using moving weights and robust calculations (*Cleveland, 1979*).

Additionally, as $K$ changed linearly over time, the linear relationship was approximated by robust line fitting (*Tukey, 1977*). Therefore, the slopes of the lines indicate the changes per day.

## RESULTS

With the first and second axes of the original PCA, the variants thus far were divided into groups of 0 to 3 (Fig. 1A). These groups were commonly observed in other countries on all continents (*Konishi, 2021c*).

Figure 1B shows how the rates of these groups changed (left axis). The green and pink backgrounds represent the period of the state of emergency and the Go To travel campaign, respectively. The same panel also shows the number of detected cases, PCR tests, and foreign visitors (right axis). The number of public PCR tests, which were given only to those with clear symptoms, was 19 times the number of confirmed cases. This testing rate is lower than that of some successful countries by one to two orders of magnitude; for comparison, on Feb 22, 2021, Australia has 39 estimated cases and 34,800 PCR tests per day (*Department of Health, Australia, 2021*). For the last 30 days from the same date, New Zealand had 802 positive results and 194,233 negatives (*Ministry of Health, New Zealand, 2021*). Singapore had 196 positives while performing 32,100 PCR

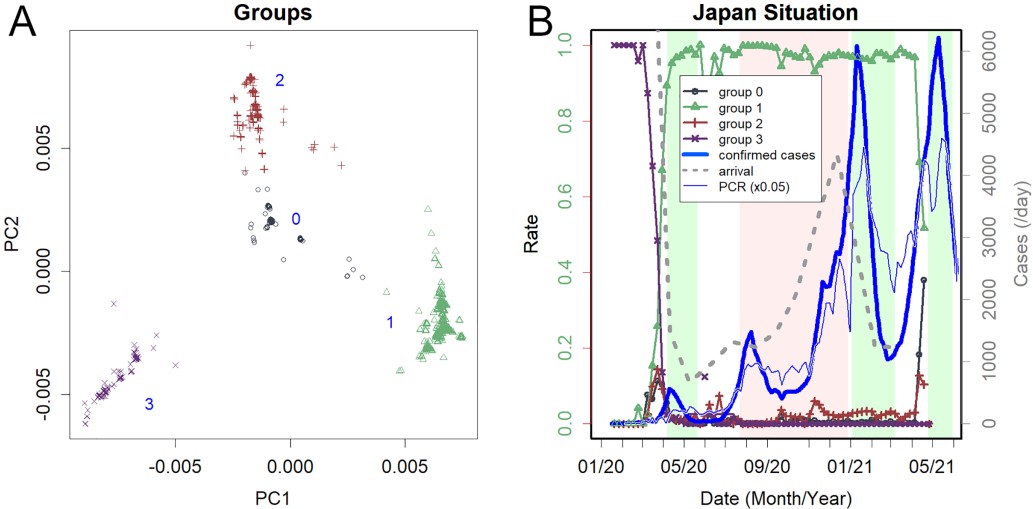

**Figure 1 Changes in groups of variants found in Japan.** (A) The scaled principal component (sPC) of the sequence found in Japan. Axes PC1 and PC2. Four groups are obvious: the clearly separated groups common to other countries (*Konishi, 2021c*). Numbers from 0 to 3 are tentatively assigned. (B) Increase/decrease of each group. Group 3 was dominant until March 2020; Group 1 then became dominant, and Group 0 was rapidly increasing in May 2021, reflecting the increase of B.1.160.1 (left axis). The green background indicates the periods of the states of emergency, and the pink background reflects the Go To travel campaign period, which is currently suspended. The thick blue line is the number of confirmed cases, and the thin blue line is 1/20 the number of PCR tests (right axis). PCR is performed only on symptomatic patients, and the total number of tests was, on average, 19 times the number of positives. The grey dotted line is the number of arriving visitors.

tests per day (*Ministry of Health, Singapore, 2021*). Iceland had only one domestic infection within 7 days but still performed one thousand tests per day (*Iceland Government, 2021*).

Until March 2020, only Group 3 variants were identified. This group includes the earliest variants reported from China (*Wu et al., 2020*; *Zhang et al., 2018*) and those found on the cruise ship *Diamond Princess*. Since then, this group has disappeared worldwide, perhaps because of lower infectivity relative to other groups (*Konishi, 2021c*).

Subsequently, Group 1 was the mainstream group throughout 2020 (Fig. 1B). It peaked in April 2020 and was eventually replaced by more mutated variants; Figs. 2A–2C show typical examples of when and how many such variants existed. It should be noted that what defines a *variant* is arbitrary. In fact, the classification here sometimes disagrees with the Pango lineage (*Rambaut et al., 2020*), which is based on phylogenetic tree analysis. For instance, most of the domestic variants were categorized as B.1.1.214, with the exception of the latest blue-green variant that was categorized as R.1. However, as Fig. 2 shows, these variants are objectively distinguished from others in that they have their own characteristic PC and mutations (Table 1). Here, a new variant was defined as one with a unique value in the PC axis and that appeared during a specific time period. The same characteristics were also observed for the variants that peaked in each country. A clear example is variant B.1.1.7, which once occupied England. Because it has many mutations, it showed PC values that were significantly different from those of the rest of the population (Fig. 2D).

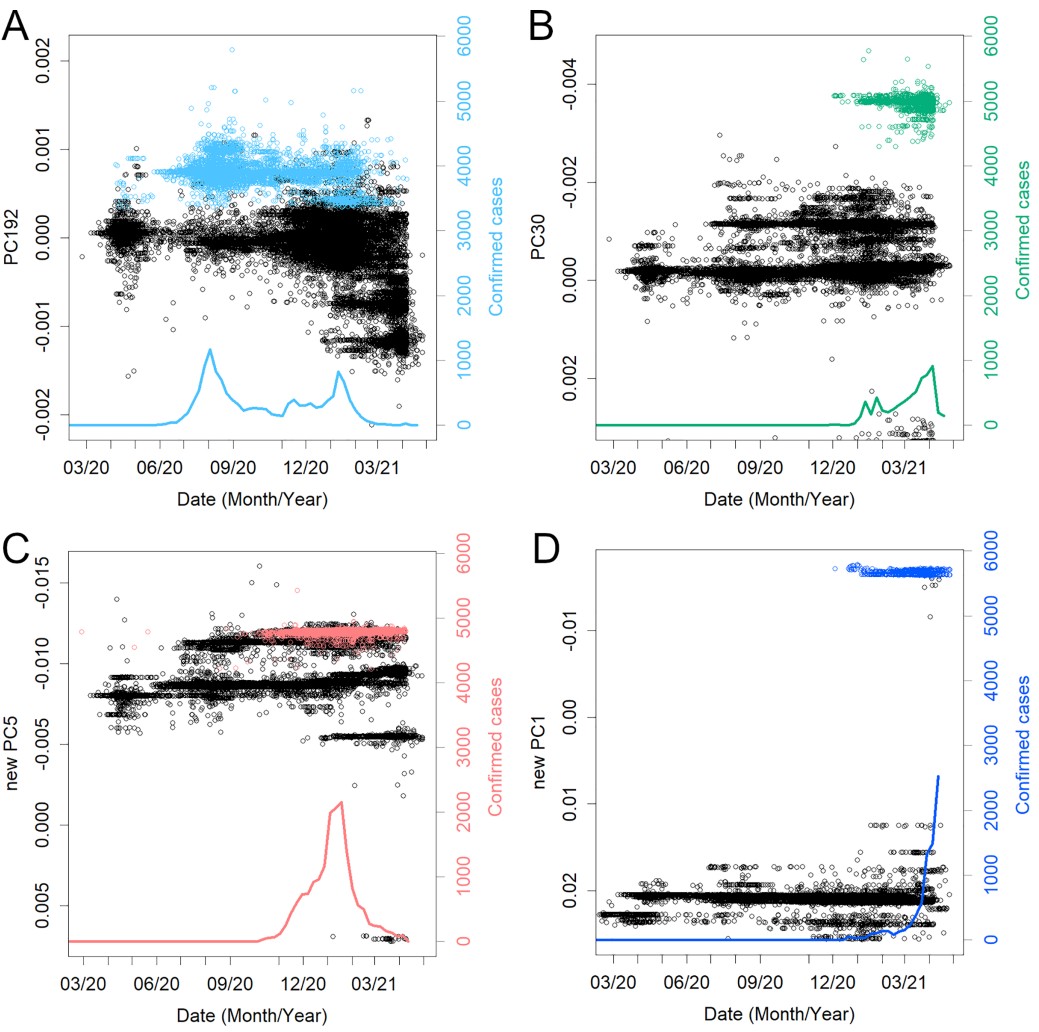

**Figure 2 Examples of some variants, time of detection, and PC values.** The colors are the same as in Fig. 4A. (A) The variant that caused the second peak. (B) The newest prevalent variant, R.1, among the domestic variants. (C) The variant that was dominant in the third peak and that also spread to South America. (D) Variant B.1.1.7, the so-called England new variant. The PC values are substantially different due to many mutations.

The number of people entering Japan was on the rise, with an average of approximately 130 thousand people per day arriving in Japan in 2019. The first state of emergency reduced this number to less than one thousand (Fig. 1B). However, the number of visitors has gradually increased since then. Visitors were asked to voluntarily quarantine in hotels for two weeks; however, as already mentioned, the inspection and quarantine of entry were not strictly enforced. Accordingly, as the number of visitors increased, overseas variants appeared in Japanese cities.

Since November 2020, variants that created the peak overseas have also appeared in Japan (Fig. 3). Variant B.1.2 was, the variant that long swept the USA and even peaked in Australia. Variant B.1.177 swept Europe. Variant B.1.160 (Fig. 3C), which was more prevalent in France during the same period, was also observed. Variants B.1.525 and

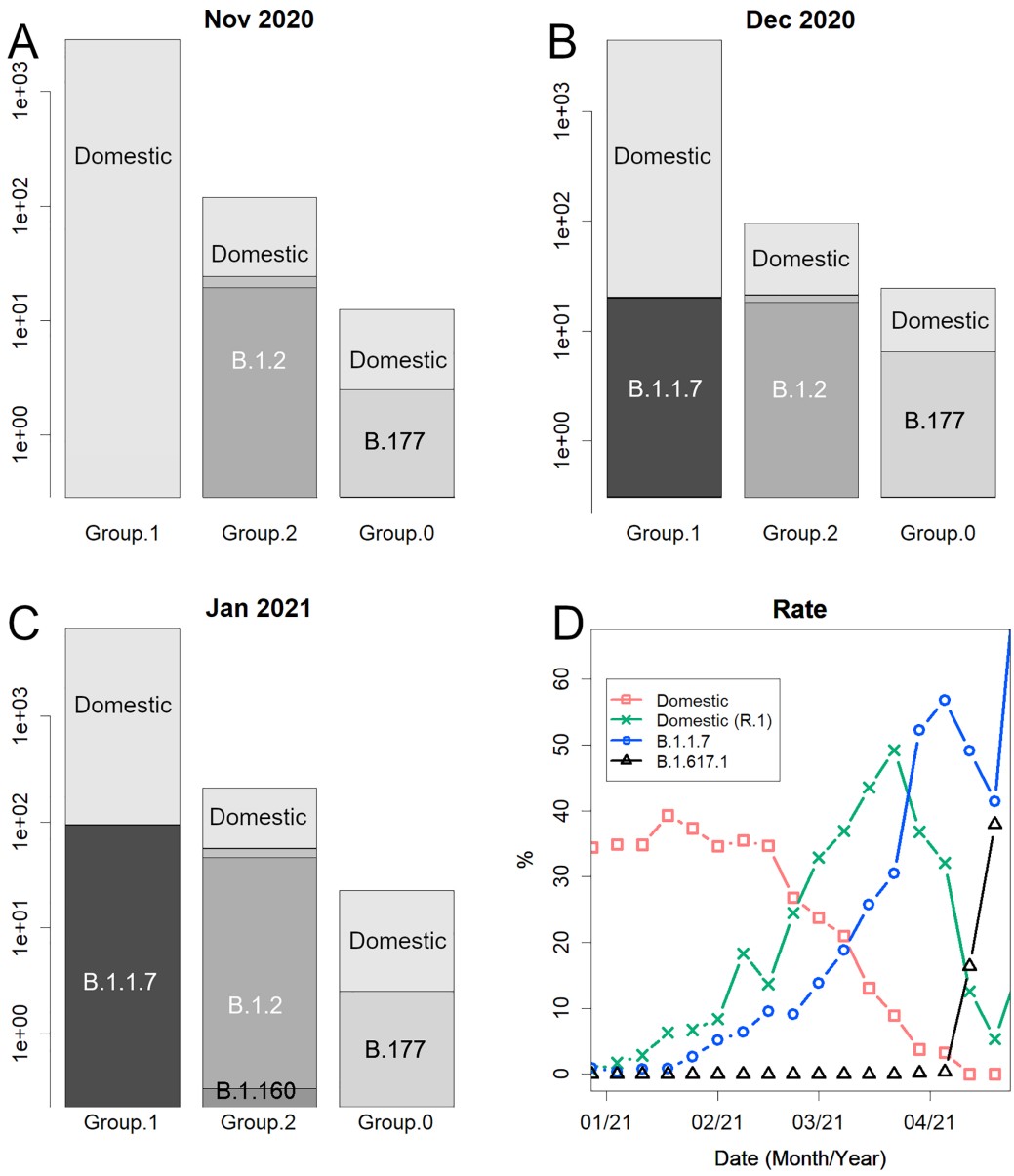

**Figure 3 Details of the number of sequences reported.** (A) Number of sequences for groups 1, 2, and 0 in November 2020, a semi-log plot. Group 1 is overwhelmingly large. Variants from overseas appeared. Variant B.1.2 was predominant in the USA. (B) December 2020. Variant B.1.1.7 appeared. Variant B. 177 swept Europe. (C) January 2021. Variant B.1.160 was predominant in France. (D) Changes in the rate of the whole in 2021. Variant B.1.617.1 was predominant in India.

B.1.1.317, which appeared in Europe, were also found, although it accounted for only in a small percentage of cases. Variant B.1.1.207, which was prevalent in North America, was also confirmed. In March, B.1.351 and P.1, which were endemic in Africa and South America, respectively, were confirmed; these should continue to be closely observed. However, in February, B.1.1.7, a variant from England, increased rapidly and became the

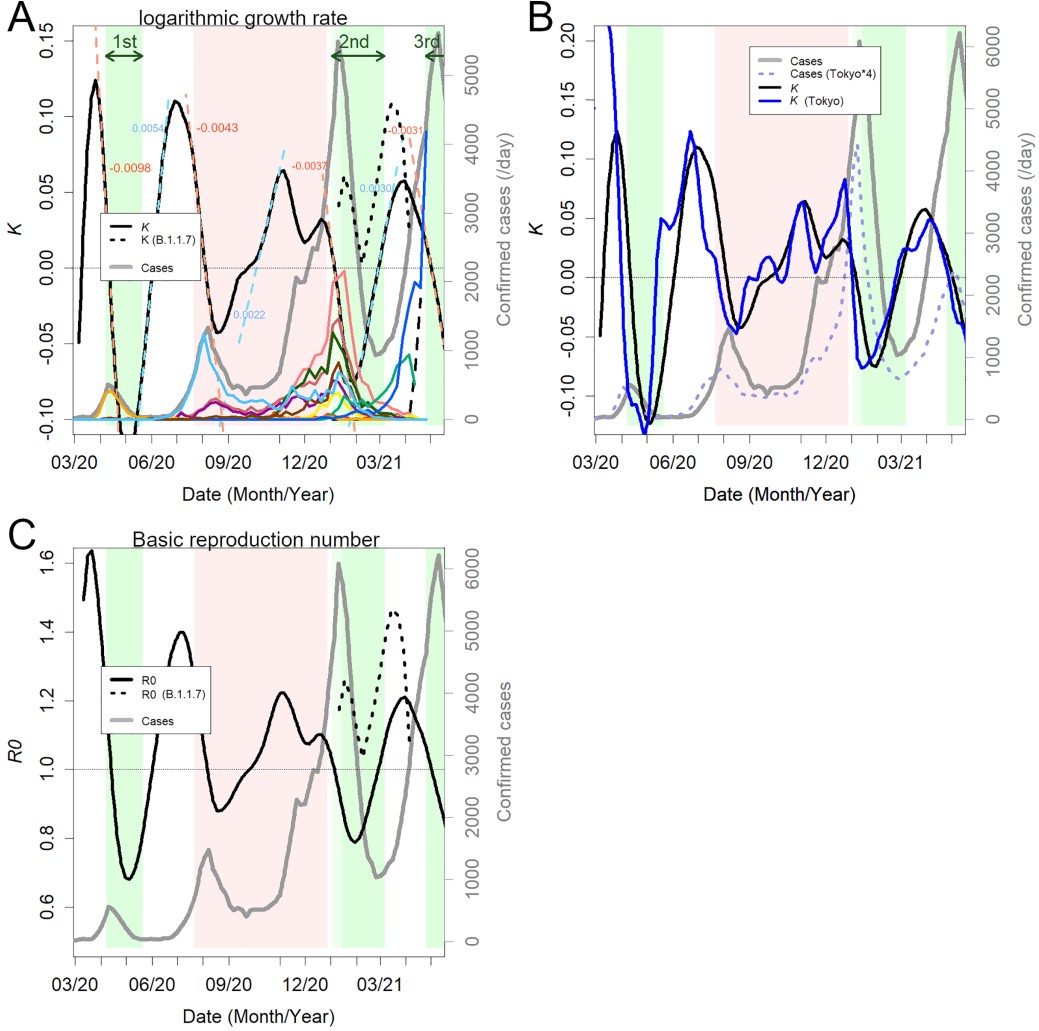

**Figure 4 Changes in the number of infected people.** (A) Logarithmic growth rate (black) estimated from the total confirmed cases. The dashed lines in light blue and orange are the approximated straight lines, and the numbers are their slopes. The decreased rates of $K$ were almost always the same. The black dotted line is the $K$ of variant B.1.1.7. The thick gray line is the number of overall cases. Colored lines are domestic variants in group 1 (Table 1). The increasing blue line is B.1.1.7. The black dashed line increasing at the end of the panel is B.1.617.1. The green background indicates the state of emergency, and the pink background reflects the period of the Go To travel campaign. (B) Comparison between the number of cases nationwide and the number of cases in Tokyo alone. The blue line is $K$ in Tokyo. (C) Estimated basic reproduction number (black), which has also changed because of the change in $K$.

predominant variant (Fig. 3D). At the end of April, B.1.617.1, which was the predominant variant in India, appeared and spread rapidly.

In Fig. 4A, we can see the response of the government to changes in the number of variants in more detail. Here, the logarithmic growth rate $K$ of the confirmed cases is shown to make it easier to understand the triggers for the increase and decrease in the number of patients. A more widely used basic reproduction number, $R_0$, is also shown (Fig. 4C); however, the calculation of this number is less objective, and the value may not

represent the true $R_0$ because it requires the selection of an infection model and the estimation of the average duration of infection (*Delamater et al., 2019*).

If the cases grow exponentially, $K$ will show a constant positive value; however, the value increased and decreased linearly (Fig. 4A). The movement is approximated by the light blue and orange dashed straight lines. The accompanying number is the slope, indicating the increase or decrease in $K$ per day. The linear relationships were stable for several weeks, and after a short transition period, they moved on to the next linear phase. $K$ increased in the valleys between the peaks of the confirmed cases. In addition, $K$ decreases in the top half of a peak. The peak apex coincides with the time when $K = 0$ during the descent of $K$. There is a delay of one to two months from the time when $K$ begins to rise or fall until there is a clear change in the number of confirmed cases. $K$ is the determinant for the number of cases; the more positive days in $K$, the more rapid the increase in confirmed cases will be.

In the first peak in April 2020, the first state of emergency was declared, but $K$ was already in the middle of a rapid decline. The variant that caused the peak, shown in tangerine, had characteristics only as of the group 1 variant; the variant disappeared immediately. Immediately after the peak, the light blue variant began to increase, and $K$ bottomed out and began to rise. However, the state of emergency was discontinued in the midst of the rising $K$.

Even when the light blue variant increased to form the second peak, a state of emergency was not declared; instead, a "Go To travel" campaign was launched instead (Fig. 4A). Despite this incoherent policy, the increase in light blue variants subsided (September 2020), but the epidemic did not stop. At least nine new variants emerged (Fig. 4A and Table 1), and their proliferation jointly kept $K$ positive, producing a sharp increase in the number of confirmed cases (Fig. 4A). The cases continued to grow during the campaign (Fig. 4A, December 2020), but $K$ began to decrease spontaneously (Fig. 4B). It should be noted that even with the highly infectious B.1.1.7 variant, $K$ decreased during this period (Fig. 4A, dotted line). Eventually, the campaign was suspended, and the second state of emergency was issued.

During the second state of emergency, $K$ began to rise again. This rise was due to the fact that two new variants began to proliferate. The blue-green variant R.1 is one of the newest domestic variants and has two mutations in its spike; another blue variant was identified as B.1.1.7 (Fig. 4A, Table 1). As a natural consequence, after the state of emergency was discontinued, $K$ continued to increase and reached a peak, and the number of infected people increased again. The third state of emergency was not issued until $K$ decreased again, and the number of infected people continued to increase. B.1.1.7 showed a higher $K$ and $R_0$ than the whole but peaked earlier; this is likely due to the increase in the next variant, B.1.617.1 (Fig. 3D).

Before the Go To Travel campaign, the change in $K$ in Tokyo was about two weeks ahead of that nationwide. However, after the campaign, the difference was almost eliminated (Fig. 4B).

The B.1.1.7 variant has been mutating since it arrived in Japan, similar to the domestic variants (Fig. 5). For a period of time, it looked as if the increase had slowed (Figs. 4A and
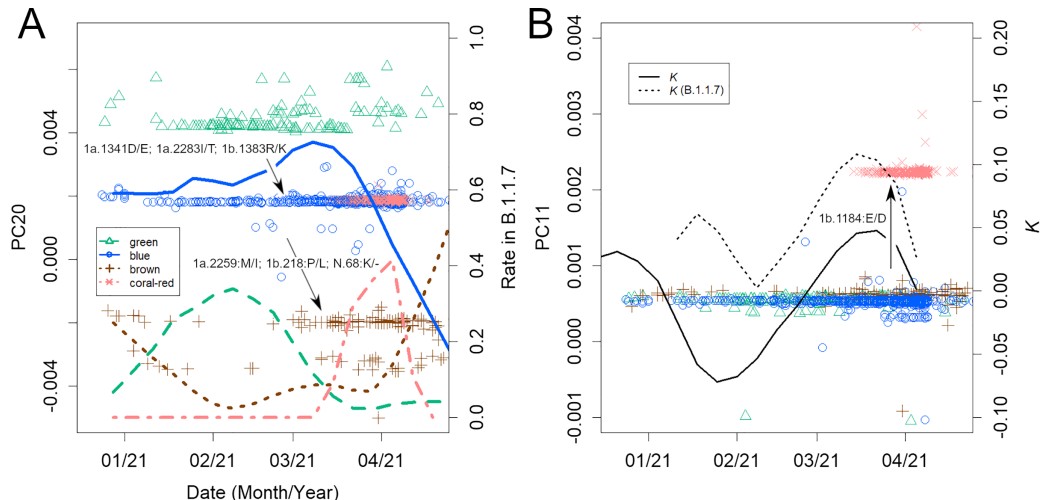

**Figure 5 Changes observed in variant B.1.1.7.** (A) The first green variant changed gradually to blue and then brown with three amino acid changes each. The dots represent sPC20 values, and the lines represent the rate of each among all B.1.1.7. (B) sPC11. One amino acid mutation from blue produced coral-red, and this difference was detected on this axis. The lines show the change in $K$. After the growth of the initial variant was suppressed, the mutated variants created the source of the next peak.

5B, dotted line). However, it has begun to increase again as of the writing of this paper, likely because of these mutations. Such mutations have been found along several axes, producing several variants.

It should be noted that the linear increase or decrease of $K$ was not affected by the states of emergency (Fig. 4A). Except for the first peak, $K$ decreased at the same rate. For the second peak, a state of emergency was not issued. For the fourth peak, one was issued after the descent of $K$. During the third peak, although the state of emergency seemed to have an effect, the rate of decrease in $K$ was the same as during the other two. This was also true when the data were selected solely for Tokyo, where the states of emergency were issued (Fig. 4B). The rate of increase of $K$ also remained constant regardless of the presence or absence of the states of emergency. If declarations alter the rate of change in $K$, then these lines should be curved.

The inadequacy of the immigration policies not only allowed powerful variants from abroad to enter the country but also led to the transfer of variants. The three major variants that formed the third peak (Fig. 4A, coral red, flesh pink, and green) spread to South America, where they accounted for about 10% of the epidemic by the end of February 2021 (Fig. S1). In addition, R.1 (Fig. 4A, blue-green) caused approximately 1,000 cases in North America in April 2021. A variant in B.1 (group 0 and new PC21 > 0.0024) appeared in January 2020. As group 0 has been rather minor, Japan may not be its origin country, but it was the only country in which an epidemic was confirmed at that time. This variant became prevalent in the USA from August to September 2020, accounting for a quarter of the total cases.

## DISCUSSION

Group 1 variants were changing in Japan in 2020, causing three peaks of the epidemic. At the end of the year, the number of variants originating overseas began to rise, leading to a large fourth wave from May to June 2021. PCA was helpful in properly identifying and classifying variants. If we had not been able to distinguish that these were many separate variants, we could not have known why $K$ was positive during the Go To travel campaign or why cases waved, forming peaks. Additionally, the variants that were transferred overseas were buried among the other variants.

Each ascent of $K$ was caused by a new variant (Figs. 2, 4A, and 5). Currently, SARS-CoV-2 is considered to be in the process of acclimatization to humans (*Konishi, 2021c*). A variant that has mutated to be more infectious would nullify the countermeasures people have been taking, and $K$ would, therefore, increase. The change did not necessarily occur in the spike protein (Table 1), which would be the target of immunity (*Harvey et al., 2021*). This shows that herd immunity has not yet been established and may be the reason why B.1.351, which has many mutations in the spike, has not caused a large epidemic. Unfortunately, the increased number of patients has increased the possibility of further mutations, creating a vicious cycle.

There is a marked difference in government measures between countries that succeeded in controlling the virus and those that did not. To maintain cleanliness, it is necessary to identify and isolate virus carriers, and a lockdown is essential until this task is completed. A sufficient number of repeated PCR tests are required for this task. Otherwise, we will not be able to control a pathogen that is growing exponentially and mutating rapidly because an epidemic can emerge from a single asymptomatic carrier (*Johansson et al., 2021*). This is a mathematical truism, but the reality is supporting it. Repeated and incomplete blockades are the fallacy equivalent of using a small piece of force instead of the utmost use of force, which has long been considered the wrong tactic (*Clausewitz, 1832*). This is because such incomplete measures only increase the economic cost without any practical benefit. Vaccination is another promising strategy, but Japan is lagging behind, and the program remains undecided (*Ministry of Health Labour & Welfare, Japan, 2021a*, *2021b*).

Japan is unique in that many variants are simultaneously prevalent. In contrast, in many other countries, one major variant, which is mutated domestically or imported from abroad, is prevalent (*Konishi, 2021c*); for example, the epidemic Group 2 variant, B.1.2, may have mutated domestically in the USA and subsequently entered Australia and Japan, causing new epidemics in those countries (Fig. 3). The earliest record of B.1.177, the pan-European variant, was from Spain, which was occupied by its parental variant. This variant has caused epidemics in many European countries. The new variants B.1.351 and B.1.617.1 may have entered South Africa and India from outside. They may have mutated in areas without sequencing, as no possible parental variants have been recorded. They all once accounted for most of the cases in the respective country. The situation in Japan is a result of the fact that the domestic variants were not contained as well as of the importation of highly infectious variants after reckless relaxation of the border.

Variant B.1.1.7 is changing (Fig. 5), similar to the domestic variants (Fig. 4A, colored lines). Variant B.167.1 will likely mutate as well, producing a new variant that is responsible for another peak.

To date, three states of emergency have been issued. These specifically recommend working remotely and prohibit eating out at night, especially the serving of alcoholic beverages. However, the number of PCR tests did not increase, and positive patients were not isolated. There are no restrictions on people's activities, and the infamous *jam-packed train* remains unchanged. Each state of emergency was untimely; they were not issued until $K$ began decreasing spontaneously, and they ended in the midst of the ascent of $K$ (Fig. 4A). In addition, the effect of the states of emergency was almost ineffective; it could not stop the increase of the next variant, and it did not alter the rates of the changes in $K$. It is clear that such half-hearted measures were completely ineffective. It is likely that the infection is spreading through non-meal-related situations, such as daily life at home, school, or work or commuting on trains.

The Go To travel campaign may have spread this virus over a wide area of Japan. Until that point, the timing of the increase and decrease of the value of $K$ in Tokyo had been around two weeks ahead of the rest of the country. It should be considered that the loss of this difference in timing is the result of people mixing together due to increased human flow (Fig. 4B). This may give the virus more chances to cause infection and hence to mutate, and it has expanded the number of confirmed cases during the third peak. The only government measure that has worked thus far may have been the cancellation of this campaign. It was not the government's declaration that lowered $K$ but, perhaps, the voluntary efforts of the people. Nevertheless, it is also true that the government has repeatedly released the information necessary for the efforts, and this should be commended.

The epidemic did not increase exponentially, but was even more explosive. The value of $K$ consistently increased or decreased (Fig. 4); such linear movement of $K$ was ubiquitously observed in other countries as well (Fig. S2). The constancy of $K$ is the basis of various mathematical models, but the actual mechanism of infection seems to be different in SARS-CoV2. The mechanism of the linear increase and decrease of $K$ is currently unknown; it seems that a novel mathematical model will be required to explain this movement. The value of $R_0$ also fluctuates; when $K$ changes linearly, $R_0$ will do so exponentially; hence, this parameter is unstable and not suitable as an indicator. Additionally, it is difficult to compare $R_0$ because of the problem of differences in the models (*Delamater et al., 2019*). It would be inappropriate to continue using $R_0$ as long as the standing mathematical models do not explain the situational reality.

The value of $K$ will be useful in predicting infection status, as the ascent or descent precedes the fluctuations in the number of cases by a month or two. This index can be easily calculated. Unlike $R_0$, this is a raw physical value, which does not require a complex model or estimation of $\tau$. Accordingly, $K$ would be a better choice when comparing results among different situations, at least at the present time.

## CONCLUSIONS

PCA sensitively detected new variants and allowed us to compare them over time. As a result, several aspects were clarified. The original intention of Japan to block the transmission route and resolve the infection was unsuccessful. Moreover, the partial lockdown that requests only self-suppression failed to contain the virus. Meanwhile, domestic variants continued to mutate and caused multiple epidemic peaks. A poorly managed waterfront operation failed, and a number of highly infectious variants were introduced into the country, while others were taken out of the country. As a result of these inappropriate measures, multiple infectious variants have spread and have taken root in Japan. Over 80% of the present variants are of foreign origin, and they seem to continue changing. If the borders are relaxed again, more new variants will emerge, and the rooted variants will be released from the country to the world.

## ACKNOWLEDGEMENTS

We would like to thank Editage for English language editing.

### Funding

The authors received no funding for this work.

### Competing Interests

The authors declare that they have no competing interests.

### Author Contributions

- Tomokazu Konishi conceived and designed the experiments, performed the experiments, analyzed the data, prepared figures and/or tables, authored or reviewed drafts of the paper, and approved the final draft.

### Data Availability

The data are available at figshare: Konishi, Tomokazu (2020): SARS-CoV-2 found in each continents.. figshare. Dataset. https://doi.org/10.6084/m9.figshare.13315628.v2.

### Supplemental Information

Supplemental information for this article can be found online at http://dx.doi.org/10.7717/peerj.12215#supplemental-information.

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
