# Peer review of "Effect of control measures on the pattern of COVID-19 Epidemics in Japan"

_PeerJ, doi:10.7717/peerj.12215_

## Round 0.1 · original submission · Major Revisions

Some major issues have been indicated by the reviewers and the authors should send a new version of the text trying to solve them. Please, see the comments below so as to have more information.

Reviewer 1 ·

Basic reporting

No comment

Experimental design

The author attempted to examine the pattern of the pandemic in Japan. However, he has achieved only partial success in achieving his objective. In fact, I have some comments, all of them major.

1.- Although the author indicates in the title 'influenced by control measures' I do not see anywhere that it evaluates or, nor even that it controls by those measures.
The author should either change the title or add sections (in methods, in results and in discussion) where he evaluates these control measures in detail.

2.- The method section is very poorly explained and is very difficult to reproduce because it is not known how he applies the methods he says it uses. Among others:
2.1.- Explain DECIPHER in some detail and not just reference.
2.2.- How did it converted to a Boolean vector?
2.3.- What role do the variables confirmed cases, PCR tests and number of foreign visitors play? Did the author include them directly along with the nucleotid sequences in the PCA?

3.- Why were those the conditions for finding variants from each country? What is this range of values based on for each of the components?

Validity of the findings

The results section is also very incomplete.

1.- The author should provide a descriptive analysis of all the variables, distinguishing between Japan and each of the countries involved in his analysis.


2.- What is the variance collected by the components? How many components were chosen? The same for Japan as for other countries?

3.- In the Discussion section, the results are only compared with Johansson et al., 2021 and with a previous work by the same author.
He should compare his findings in detail with those of other studies indicating their advantages and disadvantages.

4.- It should include a paragraph, at least, of limitations, explaining how these have affected your results and what was done to solve them or, in any case, why they could not be solved.

Additional comments

The author attempted to examine the pattern of the pandemic in Japan. However, he has achieved only partial success in achieving his objective. In fact, I have some comments, all of them major.

1.- Although the author indicates in the title 'influenced by control measures' I do not see anywhere that it evaluates or, nor even that it controls by those measures.
The author should either change the title or add sections (in methods, in results and in discussion) where he evaluates these control measures in detail.

2.- The method section is very poorly explained and is very difficult to reproduce because it is not known how he applies the methods he says it uses. Among others:
2.1.- Explain DECIPHER in some detail and not just reference.
2.2.- How did it converted to a Boolean vector?
2.3.- What role do the variables confirmed cases, PCR tests and number of foreign visitors play? Did the author include them directly along with the nucleotid sequences in the PCA?

3.- Why were those the conditions for finding variants from each country? What is this range of values based on for each of the components?

The results section is also very incomplete.
4.- The author should provide a descriptive analysis of all the variables, distinguishing between Japan and each of the countries involved in his analysis.
5.- What is the variance collected by the components? How many components were chosen? The same for Japan as for other countries?

6.- In the Discussion section, the results are only compared with Johansson et al., 2021 and with a previous work by the same author.
He should compare his findings in detail with those of other studies indicating their advantages and disadvantages.

7.- It should include a paragraph, at least, of limitations, explaining how these have affected your results and what was done to solve them or, in any case, why they could not be solved.

Reviewer 2 ·

Basic reporting

See below

Experimental design

See below

Validity of the findings

See below

Additional comments

The work is average but may be improved by the inclusion of the following suggestions.

1. English should be improved throughout the manuscript.
2. Quantitative information should be provided in the abstract.
3. The concussion should be concise and to the points indicating the application of the work.
4. The novelty of the work should be established.
5. Refs are not updated and the following ref. The addition of this may improve the quality of this manuscript.
Sci. Total Environ., 728, 138861 (2020).

Reviewer 3 ·

Basic reporting

A background on Covid-19 in Japan is covered in the Introduction. Part of the Introduction should be written clearly and concisely on what is the problem the paper would like to address and why it is an important problem to tackle. The Introduction contains the description of the approach, however, the discussion on the justification to support the approach taken. The findings of the approach are not highlighted from the Introduction as well as the contributions of the paper are omitted. The findings by the authors have practical value that merits sharing with the wider academic community, but more work needs to be done to emphasize their technical and applicability aspects as well as to more convincingly demonstrate the original contribution that this work makes to the scholarly knowledge base. Both findings and contributions should be clearly stated in the introduction part of the paper.

In terms of related work, a list of COVID-related studies and basically the related work in coronavirus invariants studies in other countries are not discussed in the paper. An important first step that will need to be taken to address this shortcoming is the inclusion of a comprehensive literature review. In its present form of the manuscript, there is no attempt to present a comparison of the related work features, strengths, and weaknesses, and more importantly, to show how the present work builds upon and extends what has already been done. Also conspicuously missing from the literature review is coverage of the relevant scientific literature on the control measures, the disease spread patterns, the mutation within the virus variants, and the intersection between these areas, especially as they relate to surveillance and prediction of risk within both infectious and non-infectious diseases.

In addition to performing the analysis with the confirmed number of cases, providing information such as the number of confirmed cases per population or the reproduction rate could enhance the interpretability of the content regarding the severity of the disease spread and the effect of the control measures at the national scale.

Experimental design

The author described the definition of PCA at the beginning of the paragraph. While cursory mention is made in the earlier section “phylogenetic trees“ and the fact that they require unverifiable assumptions, however, there is no attempt to make a discussion on the use of PCA in solving the problem at hand (among other possible methods) is not clearly justified. Background on PCA is extensively presented with the details of the method presumably coming from the earlier work.

Validity of the findings

Coronavirus is an ongoing pandemic and comparing the results with the information from one specific point in time (such as from a news website, the report from other countries figures need or from the relevant ministries website, it is important to report when the snapshot was taken precisely for validity purposes throughout the manuscript. Many factors may shape the response strategies of each country and the failure to contain the virus could be contributed by several factors other than the partial lockdown measure, insufficient PCR tests, and the relaxed border control protocol.

The findings on the changes in variant patterns and the timing should corroborate, triangulate and further confirm with the mobility patterns of visitors, and the civilians, as well as the temporal and spatial distribution of the cases in addition to the number of cases, or the number of visitors.

To make a real contribution to the field, the Discussion and conclusion will need to be substantially improved to (a) situate the scientific outcomes, findings, and lessons learned from the study in the wider context of the literature surveyed in the (expanded) literature review; (b) offer evidence-based practical recommendations for researchers, policy planners and system developers that are grounded in the researchers’ findings and experiences/observations; and (c) consider how the present work might apply to or otherwise have a value that transcends the discipline of epidemiology.

Additional comments

Line 71 - 73: “..converted to a Boolean vector, and subjected to PCA (Konishi et al. 2019). Sample principal components (PCs) were scaled based on the length of the sequence (Konishi 2015).”
The same information is duplicated in describing the PCA in the following paragraph.

Line 53: At present, over 500 victims with 54 mild symptoms are waiting at home without medication in Tokyo (Bureau of Social Walfare and Public Health 2021).
A vague word “At present” should be clarified.

Line 53: At present, over 500 victims with 54 mild symptoms are waiting at home without medication in Tokyo (Bureau of Social Walfare and Public Health 2021).
The word “patients” would be preferable to “victims”.

Line 75: “...The number of confirmed cases was obtained from the WHO...”
For the first appearance WHO should be written as “World Health Organization”.

Line 142-145: “Australia has 39 estimated cases and 34,800 PCR tests per day. For the last 30 days, New Zealand had 802 positive responses against 194,233 negatives. Singapore had 196 positives, performing 32,100 PCR tests per day. Iceland had only 145 one domestic infection within seven days, but still performed a thousand tests per day.“

The numbers of cases in Australia, New Zealand, Singapore, and Iceland are made without proper references.

Line 194: “Repeating incomplete lockdown is an error that corresponds to a piecemeal force attack.”
The phrase “piecemeal force attack” in the sentence is not easily understandable for the reader who is new to the field.

Line 214: “If the borders are loosened again..
Do you mean “If the borders are relaxed ?”

Line #2 Figure 1 caption: “Four groups are obvious; this framework is common to other countries”..
Which framework the author is referring to?

Line #2 Figure 1 caption:“Four groups are obvious; this framework is common to other countries 8 ”
Footnote 8 could not be found.

Line 243 “Ministry of Health LaW”
The reference is not cited in the main text.

Line 247 - 248: R_Core_Team. 2020. R: A language and environment for statistical computing. Vienna, Austria: R Foundation for Statistical Computing.
R Core Team is not a hyphenated word and the website needs to be provided. The reference in APA style is
R Core Team. (2016). R: A Language and Environment for Statistical Computing. Vienna, Austria. Retrieved from https://www.R-project.org/

Several references to the online resources under the References sections are not presented in a systematic way.

---

## Round 0.2 · Major Revisions

Still pending some major changes suggested by one of the reviewers. Please, try to assess them in a new revised version of the text.

Reviewer 2 ·

Basic reporting

Revision is complete and accepted.

Experimental design

Revision is complete and accepted.

Validity of the findings

Revision is complete and accepted.

Additional comments

Revision is complete and accepted.

Reviewer 3 ·

Basic reporting

-

Experimental design

-

Validity of the findings

-

Additional comments

The author(s) has addressed the reviewer's suggestions. Overall, the manuscript has been improved but the discussion section is still needs improvement.

Here are a few more detailed comments on the manuscript.
Line 53: Upon immigration -> is it upon arrival?
Line 169: Why eight groups? The author should provide justification.
Line 174: (shown in tangerine)-> missing reference Table/Fig
Line 201: (The slopes of the lines indicate the changes per day) -> missing reference Table/Fig
Line 318: customs inspection system -> immigration policies?
Line 353-356: Should not deviate from the main content of the paper. These sentences are not relevant. "This was one of the well-known mistakes that led to Japan's defeat in World War II, when fragmented forces were destroyed one after another, resulting in heavy losses (Saito 2013). Japan is again setting a bad example using the same approach."
Line 337: Each ascent of K was caused by a new variant. -> Reference to fig/Table.

The discussion section could be improved by writing succinctly in a more organized flow. Similar topics could arrange together. At present, the discussion on a topic is covered in multiple portions of the Discussion.

For example, the discussion on vaccination in this section is covered in

Line 356: Vaccination is another promising strategy, but Japan is
lagging behind, and the program remains undecided (Ministry of Health Labour and Welfare,
Japan 2021).
Line 417 - 422 Vaccination is lagging, with only 4% of the population vaccinated as of May 25, 2021 (Prime Minister's Office of Japan 2021). The rate of vaccination is around 400,000 vaccines per day; as
Japan has a population of 120 million, this situation (the outbreak of new variants, resulting
peaks in the number of patients and the chronic overflow of medical institutions) will continue
for several more years if the current measures continue. Furthermore, this prediction is only valid
if the vaccine alone is effective in preventing the disease.

---

## Round 0.3 · Minor Revisions

Still pending some minor issues to be addressed in a final revised version of the text.

Reviewer 3 ·

Basic reporting

Thank you for addressing the reviewer's suggestions. Overall, the manuscript has been improved, a few places that need to be addressed are provided below.

Line 64 - "Immigrants" -> Do you mean visitors?
Line 242 - Fig 2A - C shows -> show

Line 363 - 'To maintain cleanliness, it is necessary to identify 364 and isolate virus carriers' -> What do you mean by cleanliness?

Line 372 - 373 -> a single sentence should not be a paragraph

In the Results section, Reference figures should be added whenever the content of the figure is under discussion. The following sentences are a few examples of them.
Line 298 - "Even when the light blue variant increased to form the second peak" needs a reference figure
Line 309 - "blue-green variant R.1"
Line 310 - "Another blue variant"

Throughout the paper, the author is recommended to avoid using "very" in a sentence such as "very small" or "very difficult" and suggested using alternatives.

Experimental design

-

Validity of the findings

-

Additional comments

-

---

## Round 0.4 · accepted · Accept

All the reviewers' concerns have been correctly addressed. Therefore, I am pleased to tell you that your paper has been accepted for publication in PeerJ. Congratulations!

Reviewer 3 ·

Basic reporting

The author(s) has addressed the reviewer's suggestions. However, it is important to note that the submitted pdf file in the manuscript folder doesn't reflect the word file in the response folder. It can be easily checked with the line number 363 in pdf and 366 in the word file.

Experimental design

-

Validity of the findings

-

Additional comments

-